# A Novel Address-Matching Framework Based on Region Proposal

Yizhuo Quan [1,2], Yuanfei Chang [1], Linlin Liang [1,2] , Yanyou Qiao [1] and Chengbo Wang [1,*]

1   Aerospace Information Research Institute, Chinese Academy of Sciences, Beijing 100094, China; quanyizhuo20@mails.ucas.ac.cn (Y.Q.); changyf@aircas.ac.cn (Y.C.); lianglinlin21@mails.ucas.ac.cn (L.L.); yyqiao@irsa.ac.cn (Y.Q.)
2   University of Chinese Academy of Sciences, Beijing 100049, China
*   Correspondence: wangcb@aircas.ac.cn

**Abstract:** Geocoding is a fundamental component of geographic information science that plays a crucial role in various geographical studies and applications involving text data. Current mainstream geocoding methods fall into two categories: geodesic-grid prediction and address matching. However, the geodesic-grid-prediction method's localization accuracy is hindered by the density of grid partitioning, struggling to strike a balance between prediction accuracy and grid density. Address-matching methods mainly focus on the semantics of query text. However, they tend to ignore keyword information that can be used to distinguish candidates and introduce potential interference, which reduces matching accuracy. Inspired by the human map-usage process, we propose a two-stage address-matching approach that integrates geodesic-grid prediction and text-matching models. Initially, a multi-level text-classification model is used to generate a retrieval region proposal for an input query text. Subsequently, we search for the most relevant point of interest (POI) within the region-proposal area using a semantics-based text-retrieval model. We evaluated the proposed method using POI data from the Beijing Chaoyang District. The experimental results indicate that the proposed method provides high address-matching accuracy, increasing Recall@1 by 0.55 to 1.56 percentage points and MRR@5 by 0.54 to 1.68 percentage points.

**Keywords:** address matching; geodesic-grid prediction; pre-trained language model; attention mechanism

## 1. Introduction

Geocoding is a fundamental functional component of geographic information science that is useful for numerous applications, including map navigation [1], logistics [2], and emergency rescue [3,4]. Geocoding aims to convert location-reference text into a corresponding point or region on Earth. Early geocoding methods were mainly based on rules and templates [5], splitting a query text into address elements and then manually designing corresponding rules to match the query text with a standard address in a database. As the volume of point-of-interest (POI) data continues to grow, the expenses associated with maintaining standard address libraries are increasing significantly, and manually designed rules are struggling to handle diverse query inputs with noise. Machine-learning-based automatic geocoding techniques have attracted increasing interest from researchers [6]. Deep learning has been the most popular machine learning approach over the past decade. Deep learning can automatically extract and abstract high-level features in input text [7] without requiring the manual design of rules. Additionally, because deep learning models are exposed to a wide variety of sample data during the training process, they are more resilient to different inputs than traditional methods [8]. Therefore, recent geocoding studies have mainly focused on deep-learning-based methods [9].

Deep-learning-based geocoding methods can be divided into two categories: those that rely on geodesic-grid prediction and those that employ text–semantic address matching. Geodesic-grid-prediction methods initially divide Earth's surface into distinct regions

using a discrete geodesic grid. They then transform geocoding into a classification problem and employ a machine learning model to predict the grid affiliation of a query text. Ultimately, they use the coordinates of the selected grid's center point as the geographical location corresponding to the query text. The main advantages of such methods are their simplicity and efficiency, and ability to provide reasonable classification accuracy when fine grid scales are not required. However, these methods encounter a bottleneck in that the positional accuracy of the query text is directly constrained by the grid scale. Although a smaller spatial coverage for an individual grid would result in higher precision for text localization, a smaller spatial coverage for an individual grid implies a greater number of categories that the model must distinguish, rendering it challenging for the model to provide accurate predictions. In contrast, semantic-based address-matching methods transform geocoding into a text-matching problem in natural language processing. By comparing a query text with the textual information of POIs in a database, these methods identify the record that best matches the query description, thereby converting the query text into geographical coordinates. The main advantage of such methods is their high reliability. Once a successful match is achieved, the positioning error is typically negligible. Regardless, existing deep learning-based address-matching methods mainly focus on the semantic information in text, neglecting keyword information such as roads and landmarks, which could be used to distinguish different POIs. This may cause some candidates that are semantically similar to the query text but geographically distant to interfere with matching, thereby reducing overall matching accuracy. Intuitively, pre-filtering candidate POIs based on the keywords of a query text before matching could narrow the comparison scope, allowing a model to retain candidates that are more likely to be relevant to a query and mitigating the impact of irrelevant candidates.

Typically, when people have a specific query regarding a POI, they first identify a broad area based on the keywords in the query text and then conduct a comprehensive search within that region. This process essentially involves an attention mechanism within the human cognitive system [10], which selectively focuses on information that is more likely to be relevant to the target and excludes interference from irrelevant candidates, thereby enhancing the probability of successful matches. Inspired by this cognitive pattern, we integrated a geodesic-grid-prediction method with an address-matching model to develop a region-proposal-based address-matching framework that utilizes a geodesic-grid-prediction model to filter unrelated candidate POIs spatially. Specifically, our method consists of two stages. For an input query text, we first employ a geodesic-grid-prediction model to predict its geodesic grid in geographical space, recalling POIs located within that grid as candidates. Subsequently, we use a sentence encoder based on a pre-trained language model (PLM) to obtain semantic vector representations of the query and candidate POI texts, selecting the most relevant record based on vector similarity. We refer to the proposed approach as a region proposal based on an address-matching framework (RPAM). Experiments conducted on POI data from Beijing's Chaoyang district demonstrate that RPAM effectively improves the accuracy of existing address-matching models.

## 2. Related Work

This study was inspired by two research directions: geodesic-grid prediction and address matching. In this section, we introduce relevant research on both topics.

### 2.1. Geodesic-Grid Prediction

Wing and Baldridge [11] investigated the geolocation of texts from Wikipedia and Twitter. They proposed modeling the text-based-geolocation task as a classification problem that initially involved discretizing the Earth's surface into geodesic grids. Subsequently, a classification model was used to predict the grid cell to which an input text belonged. Geodesic grids were utilized for surface partitioning, and a statistical language model was employed to represent the probability distribution of words in documents and grids. During the prediction phase, Kullback–Leibler divergence was used to measure the distance between the

probability distributions of documents and grids. The grid with the minimum distance was then identified as the geographical location of the document. Given the extensive scope of their study and the issue of sparse data distribution in many grids, they addressed this concern in subsequent research by using a k-d tree to construct adaptively sized grids to improve performance [12]. Santos et al. [13] used machine learning techniques to perform document geolocation. They employed the LambdaMART ranking model to rank candidate place names retrieved from a knowledge base. However, this approach utilizes text-similarity calculations based on character and word frequencies without considering the semantic information within text. To address this issue, Gritta et al. [14] introduced a neural geographic encoder called CamCoder that combines lexical semantic and geographical features. CamCoder uses four input components, namely the target place name, context surrounding the target place name, other geographic entities in the text, and a sparse vector called MapVec, to represent the geographical features of locations. For vocabulary and text inputs, independent convolutional neural networks (CNNs) are employed for feature extraction, whereas for geographical features, a fully connected layer is used to map them to a dense vector representation. Subsequently, these four features undergo separate mapping using fully connected neural networks and are ultimately concatenated to form a feature vector for classification, allowing the prediction of the location to which the target belongs. CamCoder explicitly considers geographical features beyond text, enhancing document geolocation precision. However, this method relies on a geographic name dictionary as a form of external data and does not leverage the hierarchical relationships of place names on a spatial scale. To address this problem, Kulkarni et al. [15] employed S2 geometry to partition the Earth's surface. They proposed a multi-level neural geographic encoder called MLG based on a CNN to overcome the limitations of previous models.

Similar to CamCoder, MLG utilizes a CNN as a feature extractor to capture representations of the target place name, the context surrounding the target place name, and other geographic entities in the text independently. During the training phase, MLG simultaneously predicts the categories of the input text in multi-level geodesic grids, jointly optimizing the classification losses across various levels. In the inference phase, the predicted probability for each grid is the product of its own probability and that of its parent-level grid. The authors conducted a comparative analysis of MLG, CamCoder, and a single-level classification network (SLG) on three publicly available English datasets. The results indicated that MLG provides the greatest localization accuracy. Yan et al. [16] proposed a global context-embedding method to enrich the information available for geographical prediction.

Although the geodesic-grid-prediction problem has been studied extensively, previous research has primarily focused on the geolocalization of document-level texts containing target toponyms and other contextual words. The spatial scale of the grids on which these studies focused was also relatively large. For example, the average size of the target grid in [15] was 1000 km$^2$. Our research mainly focuses on predicting small-scale geodesic grids, where the average size of the target grid is 1.07 km$^2$. In this study, the entered query texts contained only the names and addresses of the target locations without any additional contextual information, which made the problem more challenging.

### 2.2. Address Matching

Address matching refers to the comparison of a query text with the textual information of POIs in a database to identify the record that corresponds to the query text, thereby transforming the query text into geographic coordinates. Existing address-matching methods are largely based on text-matching methods from the natural language processing field.

Lin et al. [17] utilized an enhanced sequential inference model to ascertain whether a query and target address matched. This method simultaneously considers the similarity of textual characters in addresses and the comprehension of semantic information within addresses. Shan et al. [18] explicitly considered the co-occurrence information between address elements, utilized Word2Vec [19] technology to obtain initial representations of nodes, and obtained the embeddings of various address elements by training a graph neu-

ral network to perform a Chinese address-matching task. Li et al. [20] proposed a multitask-learning-based approach for address matching that jointly performs address-element identification and address matching. This method incorporates the hierarchical relationships among address elements into a neural network model and introduces prior information on the hierarchical relationships of address elements through a conditional random field model. Although the aforementioned methods have enhanced text-matching models by leveraging the characteristics of addresses, these approaches require input query texts to be as complete as possible and for the address elements to be arranged in hierarchical order. However, real user queries often contain noise such as the omission of administrative division information, disordered address elements, or the use of abbreviations for place names. This noise may reduce the accuracy of the aforementioned methods, and all of the methods mentioned above utilize the Word2Vec technique to obtain vector representations of words, which are permanent and unchangeable in the problem context [21]. Consequently, these methods do not address the issue of words with distinct meanings in various contexts.

In 2018, Devlin et al. [22] introduced the bidirectional encoder representations from the transformer (BERT) PLM model, which is based on a transformer model [23]. The core of a transformer model is a self-attention operation that allows the model to generate word-vector representations dynamically by considering the contextual information of the current word. This alleviates the static representation problem encountered by Word2Vec. Most current mainstream text-matching methods use semantic retrieval frameworks based on PLMs. The essence of this approach is to convert a query text and candidate text into dense vector representations in the semantic space using PLMs. This enables text pairs with similar semantics to be closer to each other in the vector space, whereas text pairs with lower semantic similarity are farther apart in the vector space. As a result, effective differentiation among different texts can be achieved. Because this method directly models textual semantic information, it has a strong ability to generalize inputs. Based on the principles of BERT, Cui et al. introduced the Chinese whole-word mask PLM RoBERTa-wwm [24], which outperformed the original BERT model on various tasks.

To enhance the application of PLMs to address matching, the additional fine-tuning of an original PLM can be conducted. Gururangan et al. [25] proposed domain-adaptive pre-training techniques by slightly pre-training a general PLM using text from the application domain, thereby enhancing the model's accuracy for various tasks in that domain. Gao et al. [26] introduced a contrastive learning framework called SimCSE to obtain high-quality sentence embeddings. They used dropout operations to generate negative examples and updated the parameters of a PLM with triplet loss, thereby achieving state-of-the-art performance. To demonstrate the effectiveness of the proposed address-matching framework, we selected the two methods mentioned above as comparative baselines.

## 3. Methods

### 3.1. Overview

In this section, we introduce the overall structure and model details of RPAM. Our approach consists of two steps: query region proposal and semantic-similarity searching. As shown in Figure 1, when given an input query text, we first use a multi-level geodesic-grid-prediction model to predict the S2 cell to which the text belongs. Subsequently, we identify the POIs within that area as potential candidates. We then use a text encoder to obtain semantic representations of the query and candidate POI texts. By utilizing a dual-encoder architecture, we compare the query with the candidates and select the top-k records with the highest similarity as the retrieval results.

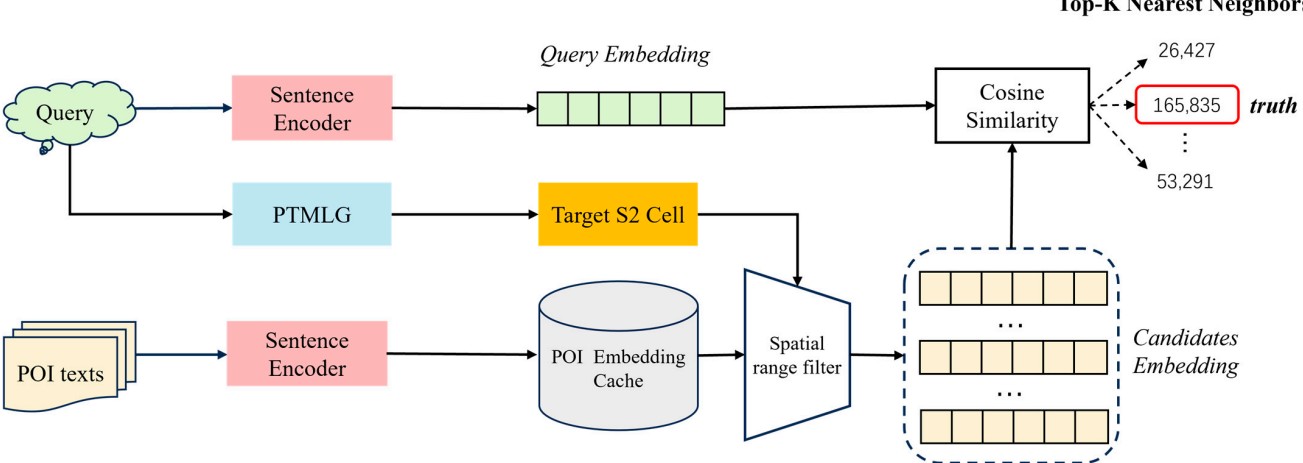

**Figure 1.** Overview of region-proposal-based address matching.

## *3.2. Query Text Region Proposal*

We modeled geodesic-grid prediction as a classification problem in line with previous grid-based geocoding research. The input is a query text and the output is the probability distribution of the input text belonging to various geodesic grids. In this section, we first introduce the multi-level partitioning of the study area using the S2 spatial index and then present a model for geodesic-grid prediction based on a PLM.

### 3.2.1. Multi-Level Partitioning Using S2 Geometry

Figure 2 presents the multi-level partitioning of the research area. We used Google's S2 geometry library (https://s2geometry.io/ (accessed on 20 April 2024)) to partition the research area into non-overlapping cells. Each S2 cell represents a category of the geodesic-grid-prediction model for forecasting. The S2 geometry is a spherical-geometry-based quadtree data structure capable of subdividing the Earth's surface into a series of contiguous quadrilateral grids. Adjacent grids at the same level do not overlap, and there is a natural hierarchical relationship between the S2 cells at adjacent levels. The S2 geometry can support grid divisions of up to 31 levels, and in our research, we selected levels 11 (Avg. 17.14 km$^2$), 12 (Avg. 4.29 km$^2$), and 13 (Avg. 1.07 km$^2$) to perform a multi-level partitioning of the research area. Level 13 is the grid scale of primary interest, whereas levels 11 and 12 are used as auxiliary predictors. Following these procedures, each level had 43, 142, and 482 output categories, respectively. Ultimately, through collaborative prediction across multiple levels, we obtained the cell indexes at level 13 to which each query text belonged.

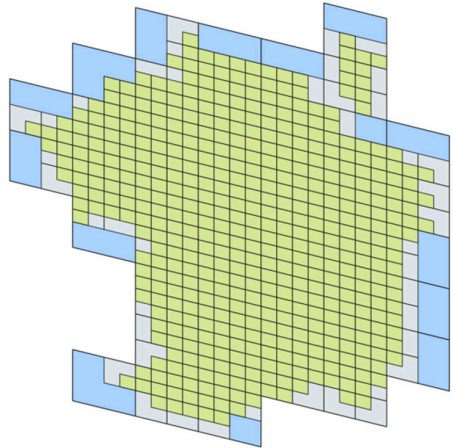

**Figure 2.** Beijing's Chaoyang district multi-level partitioning based on S2 geometry. Blue represents level 11, gray represents level 12, and green represents level 13.

### 3.2.2. Geodesic-Grid-Prediction Model

To predict the S2 cell to which a query text belongs, we developed a multi-level geodesic-grid-prediction model based on a PLM. We refer to this model as the PTMLG. In contrast to traditional single-level geodesic-grid classifiers, PTMLG considers the hierarchical relationships of input texts across multiple spatial scales. This allows for a more effective utilization of multiscale geographical information within the text. Furthermore, the prediction results at various levels can be combined by simultaneously considering the probability distribution of the query text across multiple spatial scales, which addresses the issue of reduced classification accuracy caused by the uneven distribution of POIs. The overall structure of PTMLG is presented in Figure 3a. The multi-level geodesic-grid-prediction model consists of a shared feature extractor and three identically structured classification heads. Each classification head is dedicated to predicting the geodesic grids at three different levels. Parameters are not shared among the three classification heads.

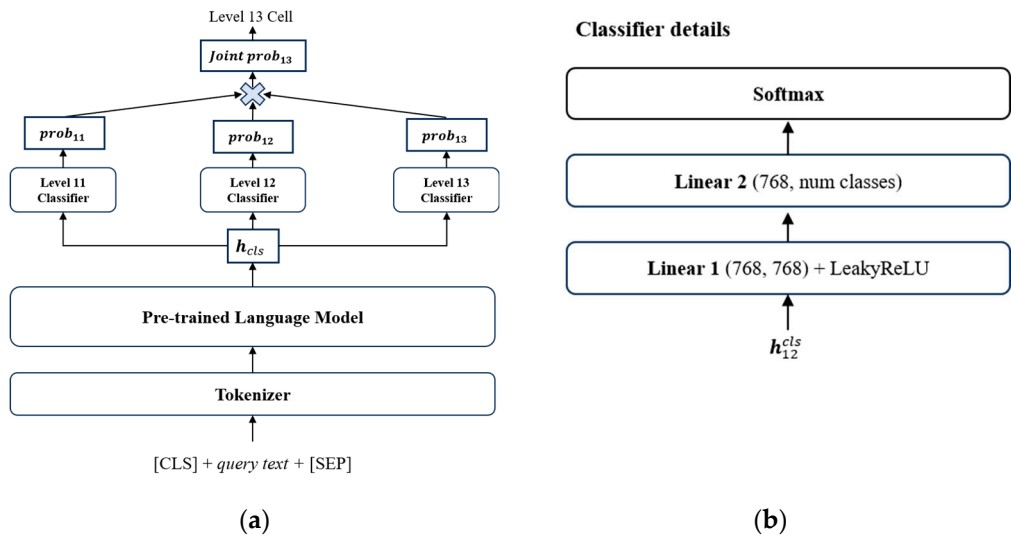

(**a**)  (**b**)

**Figure 3.** Schematic diagram of PTMLG: (**a**) overall architecture and (**b**) details of a classification head.

The PTMLG utilizes a bidirectional PLM composed of transformers to obtain deep semantic representations of input texts. To aggregate features across an entire sentence, we first add a special token [CLS] at the beginning of an input query text $S_q$ and then add a [SEP] token at the end of the sequence to mark the end. Subsequently, $S_q$ is split into $L$ tokens using a tokenizer, and each token is transformed into a dense representation $\{t_k\}_{k=1}^{L}$ using a word-embedding matrix. Considering the importance of word-position information in a sentence for semantic representation, it is necessary to obtain positional embeddings of each token in the positional-embedding matrix based on its absolute position in the sequence. Next, the dense representation $t_k$ of the token and its positional representation $p_k$ are summed and inputted into the encoder. Feature interaction is performed using a multihead self-attention mechanism that produces a semantic representation of each token in the text. This process can be represented as follows:

$$e_k = t_k + p_k \tag{1}$$

$$h_k^{(i)} = PLM(e_k) \tag{2}$$

where $e_k$ represents the input of the $k$-th token, $k \in [1, L]$, and $h_k^{(i)}$ denotes the semantic representation of the $k$-th token outputted by $i$-th layer of the *PLM* encoder. We initialized the language model's parameters using a RoBERTa-wwm-based model pre-trained on Chinese corpora. The dimensions of $e_k$ and $h_k^{(i)}$ are both 768, where $i \in [1, 12]$.

Figure 3b presents the architecture of a classification head. We use a fully connected neural network consisting of two linear layers and a softmax function to predict the probability of an input text belonging to a specific S2 cell. The LeakyReLU activation function connects the two linear layers.

We utilize the embedding of the [CLS] token outputted by the final layer of the encoder as a classification feature for the entire sentence. This feature is separately fed into the three classification heads to facilitate the prediction of different levels of S2 cells. This process is defined as follows:

$$y_L^1 = \sigma\left(W_L^1 h_{cls}^{12} + b_L^1\right) \tag{3}$$

$$p_L = softmax\left(W_L^2 y_L^1 + b_L^2\right) \tag{4}$$

where $y_L^1$ denotes the first linear layer output of the classifier of level $L$ and $p_L$ represents the probability that the query text belongs to each S2 cell of level $L$. $W_L^{(n)}$ and $b_L^{(n)}$ denote the parameters of the $n$th linear layer in the classification head of level $L$, and $\sigma$ represents the activation function.

We optimized the PTMLG parameters by minimizing the cross-entropy loss, which was calculated using the following formula:

$$L_{CE} = -\sum_{n=1}^{C} q_n log(p_n) \tag{5}$$

where $C$ represents the number of categories, $q_n$ represents the label of the $n$th category, and $p_n$ represents the predicted probability of the $n$th category.

During the training phase, we simultaneously predict the level-11, level-12, and level-13 S2 cells to which the input text belongs. The average of the cross-entropy losses at different levels is then computed to serve as the overall loss function for the entire geodesic-grid-prediction network.

$$L_{total} = w_1 L_{CE11} + w_2 L_{CE12} + w_3 L_{CE13} \tag{6}$$

Here, $L_{CE11}$, $L_{CE12}$, and $L_{CE13}$ denote the cross-entropy losses at levels 11, 12, and 13, respectively, and $w_1$, $w_2$, and $w_3$ are weight factors that are used to balance the contributions of the loss functions. Considering level 13 as the primary region and levels 11 and 12 as supporting regions, we assign weights of 0.1, 0.2, and 0.7 to levels 11, 12, and 13, respectively.

### 3.2.3. Multi-Level Joint Inference

To determine the final score for each level-13 S2 cell, we multiply the predicted probabilities of the level-13 cells by the predicted probabilities of the level-11 and level-12 cells in their respective parents. We achieve this by using the aforementioned network to determine the probability that the query text belongs to each grid at each level during the prediction phase.

$$s_{13} = p_{11} * p_{12} * p_{13} \tag{7}$$

$$\hat{y} = argmax(s_{13}) \tag{8}$$

Here, $s_{13}$ denotes the final probability that the query text belongs to each grid of level 13, and $\hat{y}$ denotes the index of the S2 cell in level 13 to which the query text belongs.

During our experiments, we noticed that some samples located at grid boundaries were mistakenly classified into neighboring cells. Because the quality of the candidate POI set recalled during the region-proposal stage directly affects the accuracy of subsequent query matching, we aimed to maximize the inclusion of true query values in the candidate set. After obtaining the prediction results for the level-13 S2 grid, we constructed a buffer zone with a 1000 m radius to serve as the final candidate region. Subsequent experiments demonstrated that this process can further improve the recall of POIs corresponding to a query.

### 3.3. Address Matching Based on Semantic Similarity

In the previous subsection, we discussed the prediction of the geographic grid to which a query text belongs to narrow down the search scope. In this subsection, we introduce natural language processing techniques for text matching to retrieve the most relevant records from candidate POIs.

To achieve efficient address matching, we utilized a Bi-Encoder architecture for address matching, as illustrated in Figure 4. The Bi-Encoder consists of two sentence encoders with shared parameters. These encoders are dedicated to obtaining sentence-vector representations for query and candidate texts, respectively. The forward inference processes of the two encoders are the same and operate independently. Similar to the geographic-grid-prediction network, we use a PLM as the underlying feature extractor to capture the semantic representations of each token in an input text.

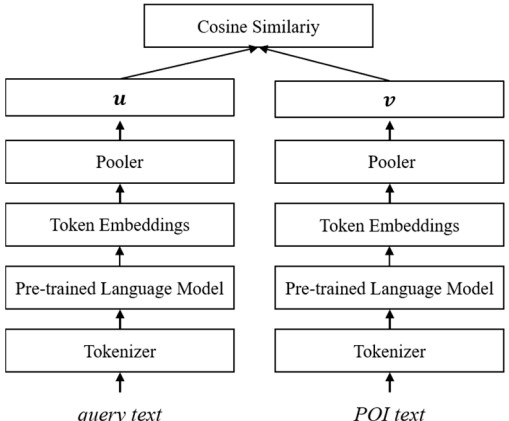

**Figure 4.** Schematic of address matching based on a Bi-Encoder architecture.

To transform an entire sentence into a fixed-size embedding for semantic retrieval, pooling operations are performed. This process yields vector representations $u$ for the query text and $v$ for the candidate POI text, which are defined as follows:

$$u = Pooler\left(\left\{h_{k,q}^{(i)}\right\}\right) \tag{9}$$

$$v = Pooler\left(\left\{h_{k,d}^{(i)}\right\}\right) \tag{10}$$

where the terms of $\left\{h_{k,q}^{(i)}\right\}$ represents the semantic representations of the query text, and the terms of $\left\{h_{k,d}^{(i)}\right\}$ represents those of the candidate POI, both of which are output by a pre-trained language model.

Finally, we use cosine similarity to measure the relevance of queries and candidates as follows:

$$sim(u, v) = \frac{u \cdot v}{\|u\| \|v\|}. \tag{11}$$

The final results of address matching are obtained by sorting all candidate POIs based on cosine similarity.

## 4. Experiments

In this section, the datasets, evaluation metrics, and implementations used in our experiments are described.

### 4.1. Datasets

To evaluate the effectiveness of the proposed method, we collected POI data from the Chaoyang District of Beijing from 2021. The Chaoyang District is one of the six primary urban areas in Beijing, covering an area of 470.8 km$^2$. The original POI data included

297,400 POIs across various categories such as dining services, residential communities, popular business districts, and road intersections. Additionally, the data included various fields such as POI names, addresses, longitudes, and latitudes. The expression forms of POI addresses vary and include door-address-type addresses, spatial-relationship-based addresses, and referential and nested address descriptions. We excluded invalid and duplicate records, removed special symbols from the POI addresses and names, and converted numbers and letters into half-width characters. After preprocessing, 287570 POIs were retained. We concatenate the names, addresses, and administrative division information of the POIs into text representations to form a reference database. The average length of the POI texts was 34.57 characters. We used 95% of the POIs to train the geodesic-grid-prediction model and reserved the remaining 5% for testing.

To simulate real query scenarios, we employed strategies such as toponym abbreviations, the removal of redundant information, and swapping address and name orders for data augmentation of the original POI texts to generate query texts. Several groups of queries and their corresponding POI texts are listed in Table 1. We manually inspected the dataset and confirmed that each query corresponds exactly to one correct POI.

**Table 1.** Examples of POI text data augmentation.

| Data Augmentation Methods | Raw POI Text | Results of Data Augmentation |
|---|---|---|
| Synonym replacement | 北京国贸商城B2层3B怪兽充电 (China World Trade Center B2 Level 3B Monster Charging) | 国贸地下2层3B怪兽充电 (Guomao Underground 2nd Floor 3B Monster Charging) |
| Element removal | 阜荣街10号首开广场F7层02-090 (Room 02-090, Floor F7, Shoukai Plaza, No. 10 Furong Street) | 首开广场7层02-090 (Room 02-090, Floor F7, Shoukai Plaza) |
| Changing address and name order | 光华路12号中信大厦 (CITIC Building, No. 12 Guanghua Road) | 中信大厦光华路12号 (No. 12 Guanghua Road, CITIC Building) |

### 4.2. Metrics

To quantitatively measure the performance of our address-matching model, we evaluate two aspects: matching performance and positional accuracy. For matching performance, we employed commonly used information-retrieval metrics, namely Recall@k [27] and mean reciprocal rank (MRR)@k [28]. Recall@k measures the proportion of queries for which the top-k retrieved results contain the corresponding POI. The query address is deemed a correct match if the corresponding POI is included within the top-k results retrieved. MRR@k represents the reciprocal of the average rank of the relevant POIs placed in the top-k retrieved results and evaluates the ranking quality of the retrieval system. For positional accuracy, we use "Accuracy@N km" [7] as the evaluation metric, which measures the percentage of predicted locations that are apart with a distance less than N km to their actual physical locations.

### 4.3. Implementation Details

All experiments were conducted using a single NVIDIA RTX 3090 GPU. We used the Pytorch (https://pytorch.org/ (accessed on 20 April 2024)) deep learning framework and transformer library (https://github.com/huggingface/transformers, (accessed on 20 April 2024)) to implement the proposed address-matching algorithm. For the geodesic-grid-prediction model, we used a batch size of 64 with the Adam optimizer and a peak learning rate of $5 \times 10^{-5}$ for 30 epochs on the training set. Additionally, we employed a linear learning rate scheduler to adjust the learning rate dynamically during the training period. This involved a warm-up phase for the first 5% of the epochs, during which the learning rate linearly increased to its peak, followed by a linear decay during the subsequent epochs. In the vector-search process, we utilized Faiss [29], a vector-similarity search library developed and made open source by a meta-fundamental artificial intelligence-research group. Faiss

enables rapid and efficient nearest-neighbor searches in vector sets of any size. Figure 5 shows a line graph of the learning rate and loss changes during training.

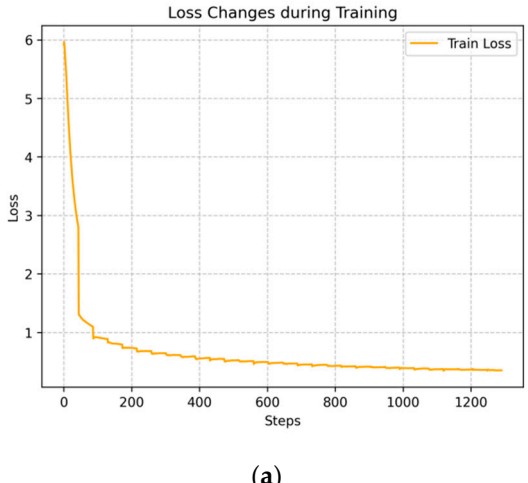

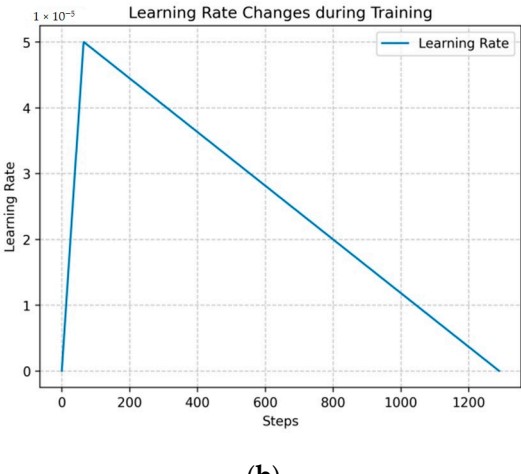

(**a**)                                               (**b**)

**Figure 5.** (**a**) Line graph of loss changes during training and (**b**) line graph of learning-rate changes during training.

## 5. Discussion

### 5.1. Effectiveness of Query Text Region Proposal

We began our investigation by examining how query text region proposals affect address-matching accuracy. We compared the matching accuracies of BERT, RoBERTa-wwm, RoBERTa-wwm-DA, and RoBERTa-wwm-CSE, with and without the proposed query text regions. BERT serves as a classic baseline for PLMs, demonstrating outstanding performance across various natural language processing tasks through the introduction of bidirectional encoders and masked pre-training tasks. RoBERTa-wwm is an improved version of BERT that employs dynamic masking strategies and whole-word masking techniques during pre-training. Domain-adaptive pre-training techniques are widely used for handling text data in specific domains. RoBERTa-wwm-DA was fine-tuned on the POI text data using domain-adaptive pre-training techniques. RoBERTa-wwm-CSE is a variation of RoBERTa-wwm trained on POI text data using the unsupervised contrastive learning technique from [26]. To ensure fair comparisons, we adopted the first–last average pooling strategy [30] for BERT, RoBERTa-wwm, and RoBERTa-wwm-DA, motivated by their demonstrated higher retrieval accuracy when employing this approach. For RoBERTa-wwm-CSE, the CLS pooling method was employed, aligning with the methodologies outlined in the original study [26]. The experimental results are presented in Table 2.

**Table 2.** Address-matching accuracies of different retrieval models with and without the PTMLG. The content in parentheses indicates the accuracy improvement after incorporating the PTMLG.

| Model | | Recall@1 | Recall@5 | Recall@10 | MRR@5 | MRR@10 |
|---|---|---|---|---|---|---|
| BERT | | 53.76 | 62.68 | 66.57 | 57.14 | 57.66 |
| RoBERTa-wwm | w/o | 60.05 | 70.97 | 75.03 | 64.18 | 64.72 |
| RoBERTa-wwm-DA | | 64.53 | 76.19 | 80.23 | 68.96 | 69.48 |
| RoBERTa-wwm-CSE | | 66.61 | 78.55 | 82.53 | 71.14 | 71.67 |
| BERT | | 55.24 (↑ 1.48) | 64.61 (↑ 1.93) | 68.67 (↑ 2.1) | 58.81 (↑ 1.67) | 59.35 (↑ 1.69) |
| RoBERTa-wwm | with PTMLG | 61.59 (↑1.54) | 72.85 (↑1.88) | 77.03 (↑ 2.00) | 65.86 (↑ 1.68) | 66.41 (↑ 1.69) |
| RoBERTa-wwm-DA | | 65.08 (↑0.55) | 76.66 (↑ 0.47) | 80.78 (↑ 0.55) | 69.50 (↑ 0.54) | 70.05 (↑ 0.57) |
| RoBERTa-wwm-CSE | | 68.17 (↑ 1.56) | 81.09 (↑2.54) | 85.43 (↑2.90) | 73.07 (↑ 1.40) | 73.66 (↑ 1.99) |

**Matching Performance.** Table 2 reveals a noticeable improvement in retrieval accuracy across various models after integrating the query text region-prediction module. Re-

call@1 increased by 0.55 to 1.81 percentage points, Recall@5 increased by 0.47 to 2.54 percentage points, and MRR@5 increased by 0.54 to 1.68 percentage points, validating the effectiveness of the matching strategy proposed in this paper, which combines query text region prediction with a semantic-retrieval model. Among all baseline models, RoBERTa-wwm-CSE exhibited the most significant improvement, with Recall@1 increasing by 1.56 percentage points and Recall@5 increasing by 2.54 percentage points. RoBERTa-wwm-DA exhibited a Recall@1 increase of 0.55 percentage points and a MRR@5 increase of 0.54 percentage points. Although the absolute improvement provided by our method may seem relatively modest, the significance of this improvement becomes meaningful when considering that the accuracy of the PTMLG is not completely accurate, indicating that there is potential for further refinement. These experimental results demonstrate that our proposed approach, which combines geographic-grid prediction and address-matching models, effectively enhances the accuracy of address matching and exhibits a certain degree of generalization across different address-matching models. Additionally, RoBERTa-wwm-CSE with PTMLG outperformed RoBERTa-wwm-CSE with a 1.99 percentage-point increase in MRR@10, indicating that leveraging the query text region-prediction model to narrow the search scope is beneficial for improving ranking performance.

**Positional Accuracy.** As shown in Table 3, integrating the query text region-prediction module also results in an improvement in positional accuracy across various models. Accuracy@0.5km increased by 1.76 to 5.71 percentage points, and Accuracy@1km increased by 2.17 to 6.67 percentage points. We attribute this to the fact that, by utilizing the geographic-grid-prediction model, we eliminate the interference from POIs that are spatially distant but semantically similar, significantly enhancing the positional accuracy of the address-matching model.

**Table 3.** Positional accuracies of different retrieval models with and without PTMLG.

| Model | | Accuracy@0.5 km | Accuracy@1 km | Accuracy@1.5 km | Accuracy@2 km |
|---|---|---|---|---|---|
| BERT | | 87.13 | 89.65 | 90.63 | 91.25 |
| RoBERTa-wwm | w/o | 88.45 | 90.75 | 91.71 | 92.28 |
| RoBERTa-wwm-DA | | 92.95 | 95.06 | 95.93 | 96.40 |
| RoBERTa-wwm-CSE | | 88.27 | 90.36 | 91.26 | 91.80 |
| BERT | | 92.84 (↑ 5.71) | 93.62 (↑ 6.67) | 97.85 (↑ 7.22) | 98.69 (↑ 7.44) |
| RoBERTa-wwm | with PTMLG | 93.21 (↑4.76) | 96.47 (↑5.72) | 98.64 (↑ 6.93) | 98.99 (↑6.71) |
| RoBERTa-wwm-DA | | 94.71 (↑1.76) | 97.23 (↑ 2.17) | 98.30 (↑ 2.37) | 98.75 (↑ 2.35) |
| RoBERTa-wwm-CSE | | 93.41 (↑ 5.14) | 96.30 (↑5.94) | 97.76 (↑6.50) | 98.60 (↑ 6.80) |

### 5.2. Explaining Why Query Text Region Proposal Is Helpful

To gain a deeper understanding of the role of query text region proposal in RPAM, we examined the top-k POIs retrieved by a single-stage semantic retriever. As shown in Table 4, we calculated the S2 cell recall for several baseline models. Successful S2 grid matching was defined as at least one of the top-k results returned by the retrieval model being within the same S2 grid as the true value. In Table 3, one can see that none of the baseline models can ensure that the top-20 retrieved results contain potentially correct results (POIs in the same grid as the query) when directly using semantic vectors for retrieval, and that retrieval results will be affected by some targets outside grid ranges. In the retrieved results of RoBERTa-wwm, 14.67% of the query top-1 results were outside the target S2 cell (the S2 cell to which the query belonged). Although RoBERTa-wwm-CSE achieves higher matching accuracy in single-stage retrieval experiments, 6.73% of the queries still have top-five results that are all located outside the target S2 cell, and 4.83% of the queries still have top-ten results that are all located outside the target S2 cell. These results indicate that the direct use of a single-stage retriever for semantic address matching may be susceptible to interference from false positives outside the grid. Region proposal for query texts

spatially eliminates interfering POIs and generates higher-quality candidates, thereby enhancing the retrieval performance of the address-matching model.

**Table 4.** S2 cell recall for single-stage address matching across various models.

| Method | Recall@1 | Recall@5 | Recall@10 | Recall@20 |
|---|---|---|---|---|
| BERT | 83.39 | 91.91 | 93.99 | 95.82 |
| RoBERTa-wwm | 85.33 | 92.73 | 94.47 | 96.08 |
| RoBERTa-wwm-DA | 89.99 | 95.75 | 97.03 | 97.94 |
| RoBERTa-wwm-CSE | 85.68 | 93.27 | 95.17 | 96.54 |

### 5.3. Effectiveness of Establishing a Buffer

Figure 6 presents the spatial distribution and heat map of the POI points that were incorrectly classified when the PTMLG model was used to predict the geodesic grid of the query texts. A darker color indicates a denser distribution of misclassified samples. Figure 5a reveals that the misclassified samples were mostly located at grid intersections. Such points near category boundaries are inherently challenging in machine learning classification problems and are prone to misclassification into adjacent grids. Therefore, we established a buffer zone based on the S2 cell-prediction results, which expands the search scope of the query text region proposal. This post-processing operation incorporates corresponding points of interest into the candidate set at the retrieval stage, specifically targeting samples that are prone to misclassification into adjacent grids, which effectively provides a high-quality set of candidates for address-matching models.

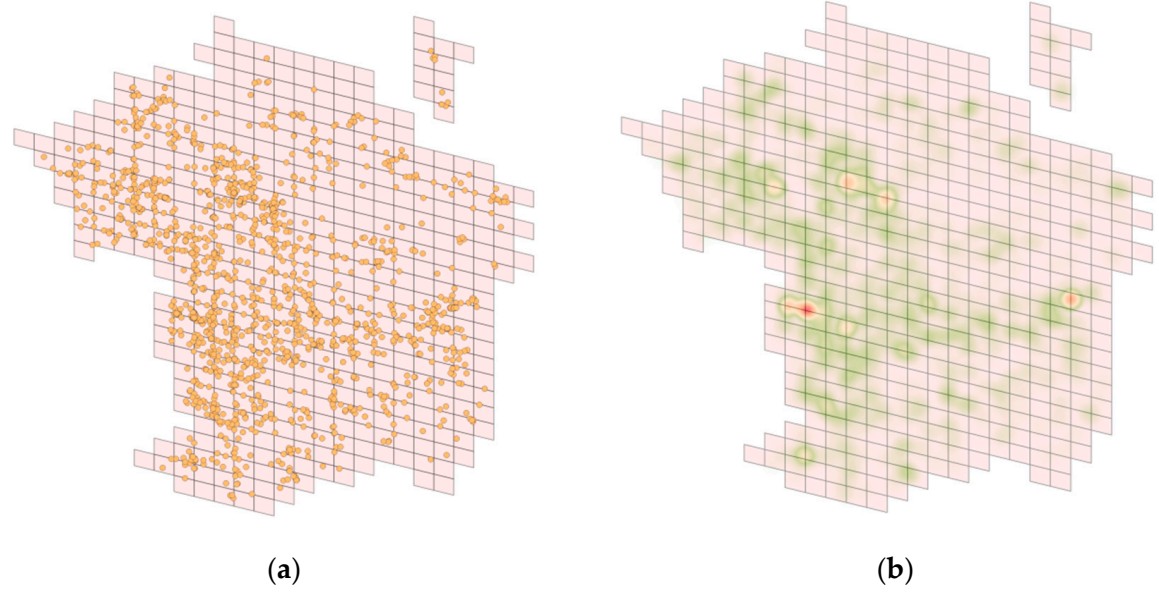

(**a**)　　　　　　　　　　　　　(**b**)

**Figure 6.** (**a**) Spatial distribution and (**b**) heat map of misclassified samples. In the heat map, red indicates a denser distribution and green indicates a more dispersed distribution.

Table 5 presents the effective retrieval rate of the geodesic-grid prediction results for different buffer-zone radii. The effective retrieval rate is the percentage of corresponding POIs that are included within the candidate set. It is evident that with an increase in the buffer-zone radius, the effective retrieval rate gradually improved. However, increasing the buffer-zone radius also introduces additional candidates, which is not conducive to maximizing the effectiveness of the query strategy in terms of narrowing the search scope. Considering that the improvement introduced by a 1200 m buffer zone is similar to that introduced by a 1000 m buffer zone, we considered a 1000 m radius to represent the optimal buffer zone.

**Table 5.** RPAM effective retrieval rate with different buffer-zone radii.

| Buffer Radius | 0 m | 100 m | 500 m | 1000 m | 1200 m |
|---|---|---|---|---|---|
| Effective retrieval rate | 89.47 | 94.91 | 97.76 | 98.52 | 98.67 |

## 6. Conclusions

In this study, we utilized the keyword information in query texts and imitated the attention mechanism used by humans when using maps, combining the geodesic-grid-prediction task and the address-matching method for geocoding. We designed a region-proposal-based address-matching framework called RPAM. RPAM initially predicts the geographical region to which a query text belongs, eliminating candidates irrelevant to the query text from a geographical perspective, thereby narrowing the search scope. Subsequently, we employed a semantics-based address-matching model to search for the POI with the highest similarity to the query text in the vector space. Through comparisons of baseline methods and in-depth experimental analyses, we demonstrated that incorporating a geodesic-grid-prediction model enhances the precision of address matching. Although the proposed framework enhances the match rate of address matching, future work is still required. Initially, the geodesic-prediction model within the proposed region proposal-based address-matching framework utilizes a fixed spatial scale for candidate regions, but this framework ineffectively manages points of varying densities. This limitation could be mitigated by implementing an adaptive discretization partition scheme [31]. Another direction for future work involves improving the embedding quality of POI text. Future research can consider developing specialized methods to enhance the distinguishability of POI text embedding. We believe that RPAM provides novel insights for tackling the address-matching problem.

**Author Contributions:** Yizhuo Quan and Chengbo Wang conceived the idea for the study, designed the proposed method, and wrote the manuscript Yizhuo Quan and Linlin Liang developed the address-matching framework and performed the experiments. Yanyou Qiao, Chengbo Wang, and Yuanfei Chang served as experts. and provided funding support. All authors have read and agreed to the published version of the manuscript.

**Funding:** This study was funded by the National Key Research and Development Program of China (No. 2022YFC3301603).

**Data Availability Statement:** The data presented in this research are available on request from the corresponding author.

**Acknowledgments:** We thank Yiming Cui, Wanxiang Che, Ting Liu, Bing Qin, and Ziqing Yang for making the Chinese RoBERT-wwm PLM available for download.

**Conflicts of Interest:** The authors declare no conflicts of interest.

## Abbreviations

The following abbreviations are used in this manuscript:

| | |
|---|---|
| POI | point of interest |
| RPAM | region-proposal-based address matching |
| PLM | pre-trained language model |
| CNN | convolutional neural network |
| BERT | bidirectional encoder representations from transformers |
| RoBERTa | robustly optimized BERT pre-training approach |

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
