# Peer review of "A Novel Address-Matching Framework Based on Region Proposal"

_ijgi, doi:10.3390/ijgi13040138_

Round 1

Reviewer 1 Report

Comments and Suggestions for Authors

The authors propose a region-proposal-based address-matching framework(RPAM) that can predicts the geographical region to which a query text belongs through combining the geodesic grid prediction and the address-matching. This work is of great significance for current geocoding and address matching, but there are some problems in the logic and details of the author's paper, the specific problems are as follows: 

Question 1:in section 3.1. Figure 1 is not refined enough. further modifications and polishing are needed. 

Question 2: in section 3.2.2 figure3. Although the author describes the specific parameter information of the model in the following text, Figure 3 requires a more detailed representation of how the entire geodesic grid prediction model represents information at different scales. 

Question 3: in section 3.2.2 formula 6. How to utilize information at different scales in the research area mentioned in 3.2.1”Level 13 is the grid scale of primary interest, whereas levels 11 and 12 are used as auxiliary predictors. Following these procedures, each level had 43, 142, and 482 output categories, respectively. Ultimately, through collaborative prediction across multiple levels, we obtained the cell indexes at level 13 to which each query text belonged.” Considering 13 as the main regions 11 and 12 as auxiliary, Formula 6 may improve accuracy by proportionally calculating the total loss value when calculating the total loss value. 

Question 4: in section 4.3, Increase the line graph of learning rate and loss changes during training. 

Question 5: in section 5. In the experimental section, only the recall rate was reflected, and the accuracy and F-value need to be further reflected. 

Question 6: in section 6. In the conclusion section, further elaboration is needed on the description of future work. 

Question 7: Suggest citing papers based on POI for address segmentation: Spatial Context-Based Local Toponym Extraction and Chinese Textual Address Segmentation from Urban POI Data.

Comments on the Quality of English Language

none.

Author Response

Dear Reviewer,

We would like to thank you for your careful reading, helpful comments, and constructive suggestions, which has significantly improved the presentation of our manuscript. We have carefully considered all comments from the you and revised our manuscript accordingly. In the following pdf file, we summarize our responses to each comment . We hope our revised manuscript can be accepted for publication.

Once again, we appreciate your valuable feedback, which will undoubtedly contribute to the improvement of our manuscript.

Thank you for your time and consideration.

Sincerely,

The Authors,

12 April, 2024

Reviewer 2 Report

Comments and Suggestions for Authors

The research introduces a geocoding model that combines the geodesic grid model with an address matching process. While I generally agree with the potential applicability of this model, I find its contribution lacking. A significant assumption of the two-stage address matching process is that all candidates are equally similar to the input address, and that the correct address is typically located within a dense grid. However, this assumption doesn't hold true when the correct address is situated in a grid with few Points of Interest (POIs). Additionally, geocoding quality is often assessed based on match rate and positional accuracy, rather than 'Recall'. The proposed method also lacks a comprehensive comparison with other geocoding methods integrated into ArcGIS, such as Dual Ranges, One Range, Single House, ZIP 5-Digit, among others. It's crucial for the proposed method to clearly demonstrate why it could outperform current models. Lastly, the language in the paper would benefit from revision by a native English speaker.

Comments on the Quality of English Language

Extensive editing of English language required.

Author Response

Dear Reviewer,

Thank you very much for your careful reading, professional advice, which has significantly improved the presentation of our manuscript. We have tried our best to improve and made corrected modifications in the manuscript. Additionally, we enlisted the services of Editage, a professional English editing company, to ensure the manuscript's language is of the highest standard. In the following pdf, we summarize our responses to each comment. It is our hope that the revisions meet your approval and that our manuscript will be considered suitable for publication.

Once again, we appreciate your valuable feedback, which will undoubtedly contribute to the improvement of our manuscript.

Thank you for your time and consideration.

Sincerely,

The Authors,

12 April, 2024

Reviewer 3 Report

Comments and Suggestions for Authors

Review of the manuscript (article) titled A Novel Address-Matching Framework based on Region Proposal

Below are some suggestions for improving your article.

·         In line 6 miss e-mail of autor (Yanyou Qiao).

·         In line 8 should be added (C.W.).

·         In line 158 should be written „Shan et al. [17] explicitly considered ...“ instead of „Shan [17]explicitly considered ...“ (several authors)

·         In lines 177–178, instead of one sentence, there should be two sentences, because part of the sentence „... In 2018, Devlin et al. „ refers to the paper [22], and not to another paper [21].

·         In line 193 should be written Gururangan et al. [24] proposed ...“ instead of Suchin et al. [24] proposed ...“

·         In line 519 should be deleted DOI:doi.

·         In lines 520–521 should be written „Hu, X.; Zhou, Z.; Li, H.; Hu, Y.; Gu, F.; Kersten, J.; Fan, H.; Klan, F. Location Reference Recognition From Texts. A Survey and Comparison. arXiv 2022. DOI: 10.48550/arXiv.2207.01683.instead of „Hu, X.; Zhou, Z.; Li, H.; Hu, Y.; Gu, F.; Kersten, J.; Fan, H.; Klan, F. Location Reference Recognition From Texts. A Survey and Comparison 2022.“

·         In lines 525–526 should be written „Lai, Q.; Khan, S.; Nie, Y.; Shen, J.; Sun, H.; Shao, L. Understanding More about Human and Machine Attention in Deep Neural Networks. arXiv 2020. DOI: 10.48550/arXiv.1906.08764.instead of „Lai, Q.; Khan, S.; Nie, Y.; Shen, J.; Sun, H.; Shao, L. Understanding More about Human and Machine Attention in Deep Neural Networks 2020.

·         In lines 547 should be written „Mikolov, T.; Chen, K.; Corrado, G.; Dean, J. Efficient Estimation of Word Representations in Vector Space. arXiv 2013. DOI: 10.48550/arXiv.1301.3781.“ instead of „Mikolov, T.; Chen, K.; Corrado, G.; Dean, J. Efficient Estimation of Word Representations in Vector Space 2013.

·         In lines 553–554 should be written „Vaswani, A.; Shazeer, N.; Parmar, N.; Uszkoreit, J.; Jones, L.; Gomez, A.N.; Kaiser, L.; Polosukhin, I. Attention Is All You Need. arXiv 2023. DOI:10.48550/arXiv.1706.03762.instead of „Vaswani, A.; Shazeer, N.; Parmar, N.; Uszkoreit, J.; Jones, L.; Gomez, A.N.; Kaiser, L.; Polosukhin, I. Attention Is All You Need 2023.

·         In lines 555–556 should be written „Devlin, J.; Chang, M.-W.; Lee, K.; Toutanova, K. BERT: Pretraining of Deep Bidirectional Transformers for Language Understanding arXiv 2019. DOI:10.48550/arXiv.1810.04805.instead of „Devlin, J.; Chang, M.-W.; Lee, K.; Toutanova, K. BERT: Pretraining of Deep Bidirectional Transformers for Language Understanding 2019.

·         In line 562 should be written „Gao, T.; Yao, X.; Chen, D. SimCSE: Simple Contrastive Learning of Sentence Embeddings. arXiv 2022. DOI: 10.48550/arXiv.2104.08821.instead of „Gao, T.; Yao, X.; Chen, D. SimCSE: Simple Contrastive Learning of Sentence Embeddings 2022.

·         In line 567 should be written „Zhao, W.X.; Liu, J.; Ren, R.; Wen, J.-R. Dense Text Retrieval Based on Pretrained Language Models: A Survey. arXiv 2022. DOI: 10.48550/arXiv.2211.14876.instead of „Zhao, W.X.; Liu, J.; Ren, R.; Wen, J.-R. Dense Text Retrieval Based on Pretrained Language Models: A Survey 2022.

·         In lines 568–569 should be written „Minaee, S.; Kalchbrenner, N.; Cambria, E.; Nikzad, N.; Chenaghlu, M.; Gao, J. Deep Learning Based Text Classification: A Comprehensive Review. arXiv 2021. DOI:10.48550/arXiv.2004.03705.instead of „Minaee, S.; Kalchbrenner, N.; Cambria, E.; Nikzad, N.; Chenaghlu, M.; Gao, J. Deep Learning Based Text Classification: A Comprehensive Review 2021.

·         In line 572 should be written „Li, B.; Zhou, H.; He, J.;Wang, M.; Yang, Y.; Li, L. On the Sentence Embeddings from Pretrained Language Models. arXiv 2020. DOI: 10.48550/arXiv.2011.05864.instead of „Li, B.; Zhou, H.; He, J.;Wang, M.; Yang, Y.; Li, L. On the Sentence Embeddings from Pretrained Language Models 2020.

Author Response

Dear Reviewer,

We would like to thank you for your careful reading, helpful comments, and constructive suggestions, which has significantly improved the presentation of our manuscript. We have carefully considered all comments and revised our manuscript accordingly. In the following pdf, we summarize our responses to each comment from the reviewers. We hope our revised manuscript can be accepted for publication.

Round 2

Reviewer 2 Report

Comments and Suggestions for Authors

NO